# DyG²T: Modeling Object Dynamics with 3D Gaussian Temporal-Spatial Particle Graph Transformer

## Abstract

Increasing interaction demands with dynamic objects require accurate modeling of their dynamics and precise prediction of motion trajectories from limited observations. Existing approaches rely on the coordinates of downsampled Key Points as the feature basis and model their interactions within local neighborhoods, resulting in the loss of fine-grained details and homogenized particle representations. In this work, we propose DyG²T, a dynamics modeling framework that leverages spatiotemporally completed particle representations for multi-scale force propagation. Spatially, each Key Point enriches fine-grained edge features and spatial geometry by aggregating position information from corresponding raw particles and relative coordinates from neighboring Key Points. Temporally, after supplementing Key Points with inter-frame relative motion offsets via Motion Align Net, the Temporal Attention is applied to aggregate Key Point features across adjacent frames, preserving the dynamic evolution patterns of particles. For comprehensive interactive modeling, a Particle Graph Transformer establishes multi-scale force propagation paths from contact-near to distant Key Points, preserving discriminative long-range dependencies critical for accurate trajectory modeling. Experiments on synthetic and real-world datasets demonstrate that DyG²T achieves accurate trajectory decoding, strong cross-object and real-world generalization.

## 1 Introduction

Recently, artificial intelligence has evolved from disembodied intelligence operating independently of the physical world toward embodied systems that require interaction with more realistic, complex real-world environments Li et al. (2024a); Zhang et al. (2025b). Against this evolutionary backdrop, real-time and accurate interaction between intelligent agents and various objects has become a fundamental prerequisite, which necessitates that the agents can infer object dynamics and predict motion trajectories based on limited observations. Consequently, object dynamics modeling has emerged as a core technical demand, aiming to capture the motion patterns of highly maneuverable objects in continuous dynamic spaces by modeling complex physical processes (e.g., force propagation Zhong et al. (2024)), with the ultimate goal of enabling precise object motion trajectory prediction to support seamless agent-environment interaction.

To acquire the object dynamics, most existing approaches Li et al. (2019b); Zhang et al. (2025a) begin by employing 3D reconstruction Mildenhall et al. (2021) to convert 2D visual observations into 3D raw particle representations of moving objects, followed by downsampling such as Farthest Point Sampling (FPS) Qi et al. (2017b) to extract sparse Key Points that serve as representatives for dynamics modeling. Broadly, the dynamics modeling methods can be categorized into two main groups. Physics-engine-based methods Li et al. (2023); Arnavaz et al. (2023) rely on predefined material equations to derive the state evolution of Key Points, but they suffer from limited adaptability when generalizing across objects with diverse materials. In contrast, differentiable network-based approaches Zhong et al. (2024) construct deep models to capture local interaction patterns within the neighborhoods of Key Points and infer their future positions, freeing the modeling process from displacement calculations based on physical equations. Nevertheless, relying on Key Points information overlooks the rich appearance details encoded in the raw particles, while restricting modeling

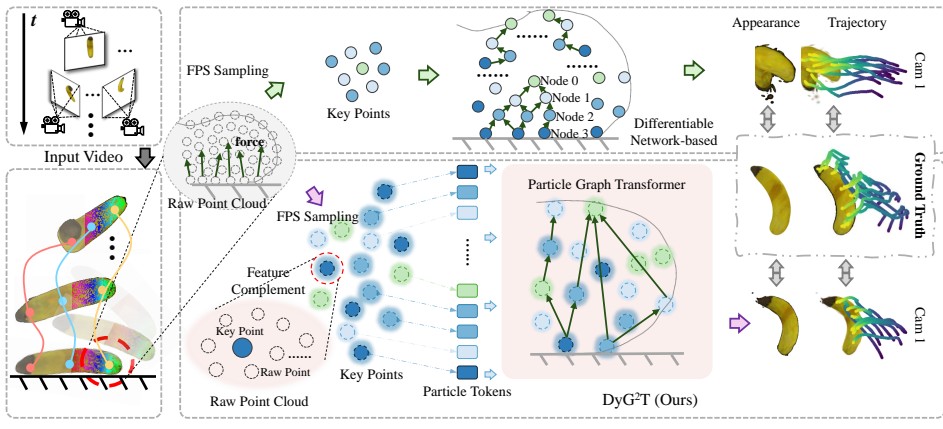

Figure 1: DyG²T vs Differentiable Network-based methods. DyG²T captures multi-scale force propagation, preserving accurate appearance and trajectories.

to spatially confined interactions tends to yield homogenized feature representations. These limitations manifest as inaccurate appearance predictions and drifting trajectories.

Specifically, in one respect, researchers adopt the coordinates of downsampled Key Points as the initial particle representation for dynamics modeling. As shown in Figure 1, FPS discards a substantial portion of raw particles in the 3D raw particle representation (*i.e.*, Raw Point Cloud) that are critical for capturing fine-grained edge details. Moreover, relying on coordinates as initial features further weakens the model's perception of object geometry and appearance. This dual loss of edge details and geometric cues compromises the integrity of the feature basis, impeding dynamics modeling and leading to errors in both appearance and trajectory decoding. Therefore, retaining the particle representation with complete and rich information is fundamental to robust dynamics modeling.

In another respect, local modeling overlooks the influence of forces across multiple spatial scales on particle representations. As shown in Figure 1, each node can only indirectly acquire multi-scale information by iterative neighborhood aggregation. This neighborhood-based update and outward diffusion mechanism tends to homogenize particle representations Sun et al. (2022). Such homogenization blurs the distinctions among forces of varying scales, ultimately causing positional decoding errors and trajectory deviations. Consequently, capturing discriminative particle representations through multi-scale force propagation modeling is also crucial for accurate dynamics modeling.

In this work, we propose DyG²T, which employs spatiotemporally completed particle representations to support multi-scale force propagation modeling. As shown in Figure 1, DyG²T leverages dynamic reconstruction (e.g., Dyn3DGS) to extract trackable particle representations from sparse-view videos. To ensure the integrity of spatiotemporal representations, at the spatial representation level, each Key Point aggregates position information from the Raw Points it corresponds to before downsampling—enriching fine-grained edge representation—and concurrently aggregates relative position information with neighboring Key Points, which enhances the ability to perceive spatial geometry. In addition, at the temporal representation level, we calculate the relative motion offset of each particle between adjacent frames using Motion Align Net and employ the Temporal Attention to aggregate particle representations across frames, enabling the retention of all Key Points' dynamic evolution patterns. For multi-scale force propagation modeling, we introduce a Particle Graph Transformer that establishes propagation paths from contact-near Key Points to those across multi-scales, preserving discriminative long-range features essential for effectively depicting the temporal collaborative evolution trajectories of all Key Points. We evaluate DyG²T on the Spring-Gaus synthetic & real-world Zhong et al. (2024) and our Unity3D-Heterogeneous datasets, demonstrating its strong cross-object generalization and real-world scalability. The contributions of this work are as follows:

- We propose a dynamics modeling framework named DyG²T, which mitigates trajectory prediction bias by preserving differentiated force propagation information within particle representations.

- DyG²T enriches the particle representations with fine-grained spatial semantics and temporal evolution features through spatial completion and temporal aggregation, respectively.

- Extensive experiments on both synthetic and real-world datasets demonstrate that DyG²T accurately models object dynamics across diverse geometries and material properties.

## 2 RELATED WORK

### 2.1 DYNAMIC 3D SCENE RECONSTRUCTION

Owing to advances in dynamic neural radiance fields Fridovich-Keil et al. (2023); Pumarola et al. (2021) and dynamic 3D Gaussians Li et al. (2024b); Lin et al. (2024), significant progress has been made in reconstructing dynamic 3D scene representations from observations. Dyn3DGS Luiten et al. (2024) achieves dense tracking of dynamic scenes on a unified point cloud. Compared with these approaches, which do not consider physical modeling, we build upon dynamic reconstruction and introduce dynamics modeling to enable future motion prediction.

### 2.2 PHYSICS-BASED DYNAMICS MODELING AND REASONING

Given the complexity of continuous motion spaces, inferring dynamics from sparsely observed initial states remains a formidable challenge. Some studies Xie et al. (2024); Zhang et al. (2024) leverage prior-configured physics engines to model material-specific dynamics, enabling reasoning over object deformations Arnavaz et al. (2023) or physical properties Qiao et al. (2022). Others employ GNNs as differentiable simulators to capture particle-level motion characteristics, encouraging progress across diverse material settings Li et al. (2019b); Zhong et al. (2024). Additionally, some researchers Xie et al. (2025) have incorporated physical constraints into interactive simulations to facilitate urban navigation. Most relevant to our work, GS-Dynamics Zhang et al. (2025a) remains constrained by GNN-based simulation, often overlooking the heterogeneity of force propagation. In contrast, our approach explicitly differentiates forces across distance scales, thereby achieving more accurate trajectory prediction.

## 3 METHOD

In this section, we introduce the details of DyG$^2$T. For each dynamic object, given a set of 2D observations $\{I_{o,t}\}_{o=1,t=1}^{O,T}$ consisting of $O$ views and $T$ frames, the goal is to predict the object's future position through dynamics modeling. As shown in Figure 2(a), we first acquire the trackable raw particle-based representations $G_t$ of the moving object from observations $I$ via dynamic reconstruction. Next, we extract Key Point $G_t^*$ using FPS to serve as the basis for dynamics modeling. Subsequently, the Spatial-Temporal Feature Completion and Aggregation mechanism operates on Raw & Key Points to perform particle-level spatial semantic completion and object-level dynamic temporal aggregation, resulting in enriched Key Point Embeddings $X_{\text{Ag}}$ (Figure 2(b)). We then employ the Particle Graph Transformer to construct direct force propagation pathways between Key Points, enabling accurate trajectory modeling and precise displacement prediction $M^{*,t}$. Afterwards, the positions of the Key Points $\hat{G}_{t+1}^*$ at frame $t+1$ is updated by adding displacement $M^{*,t}$ to $G_t^*$. (as shown in Figure 2(c)). Finally, with the aid of Linear Blend Skinning (LBS) Sumner et al. (2007); Huang et al. (2024), we interpolate the positions of Key Points to estimate the final next-frame raw particle positions $\hat{G}_{t+1}$, thus obtaining the object's future motion.

### 3.1 TRACKABLE PARTICLE-BASED REPRESENTATION BY DYNAMIC RECONSTRUCTION

Following Dyn3DGS Luiten et al. (2024), we reconstruct the trackable raw particle position sequences $\{G_1, \ldots, G_T\}$ from visual observations $\{I_{o,t}\}_{o=1,t=1}^{O,T}$, where $G_t = \{\mu_i^t | i \in [1, N]\}$, $\mu_i^t = (x_i^t, y_i^t, z_i^t)$ is the $i$-th raw particle coordinates at frame $t$. Specifically, we begin with performing static 3D Gaussian Splatting Kerbl et al. (2023) on the first-frame $\{I_{o,1}\}_{o=1}^{O}$ to obtain the particle-based appearance descriptors (*e.g.*, size, color, opacity) and the initial spatial representations, including the positions $G_1$ and rotations $r_1$. The rotation $r_1$ is represented by quaternion $\{(qw_i^1, qx_i^1, qy_i^1, qz_i^1) \mid 1 \leq i \leq N\}$, where $qw_i$ is the real part, $(qx_i, qy_i, qz_i)$ is the imaginary parts. Then, we freeze the particle appearance descriptors and adopt a recursive optimization paradigm, where each frame's particle positions and rotations $(G_t, r_t)$ are initialized from the previous frame and optimized sequentially under constraints from the observations $\{I_{o,t}\}_{o=1,t=2}^{O,T}$. This frame-by-frame optimization yields one-to-one correspondence of particles across frames, resulting in a temporal sequence of particle positions and rotations. Since rotations can be estimated via LBS, DyG$^2$T focuses on modeling the particle positions $G_t$, *i.e.*, the Raw Point Cloud.

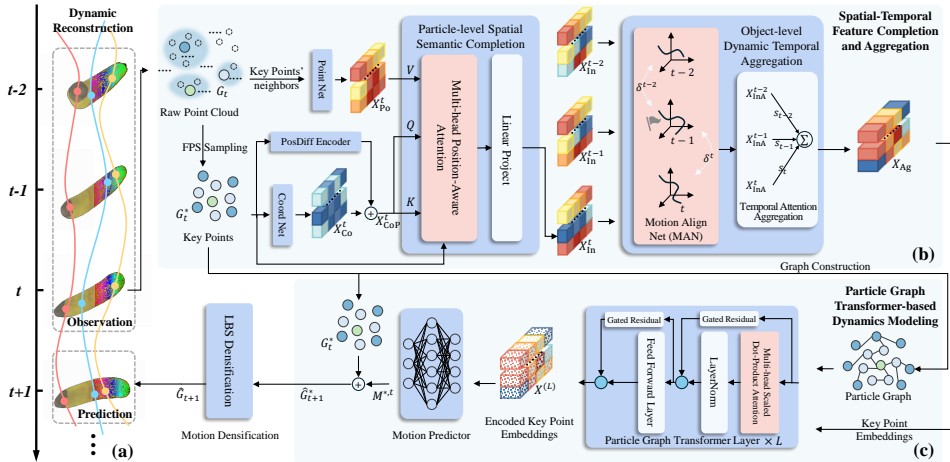

Figure 2: Overview of the DyG$^2$T. (a) DyG$^2$T utilizes dynamic reconstruction to extract trackable particle representations. (b) The Spatial-Temporal Feature Completion and Aggregation performs semantic completion and temporal aggregation at both particle and object levels, which spatiotemporally complete particle representations. (c) With the Particle Graph Transformer's global interaction modeling, DyG$^2$T captures multi-scale force propagation across the Particle Graph.

## 3.2 SPATIAL-TEMPORAL FEATURE COMPLETION AND AGGREGATION

In this section, we strengthen particle representations through two complementary perspectives. We first employ FPS Qi et al. (2017b) to extract Key Point $\mu_i^{*,t}$ from the Raw Point Cloud at each frame $t$. FPS operates by iteratively selecting the point farthest from the already selected set, ensuring a spatially uniform coverage. The resulting Key Points set $G_t^* = \{\mu_i^{*,t} \mid i \in [1, N^*]\}$ at different frame $t$ forms a sequence $G_1^*, \ldots, G_T^*$. As shown in Figure 2(b), we introduce a Particle-level Spatial Semantic Completion module that enriches the Key Points' spatial features by integrating fine-grained edge information from the Raw Point Cloud and relative positions between Key Points. Subsequently, the Object-level Dynamic Temporal Aggregation module employs a Motion Align Net (MAN) to compensate for inter-frame motion offsets, and aggregates particle features across adjacent frames (*i.e.*, $t-2$ to $t$), enhancing object-level dynamics perception.

### 3.2.1 PARTICLE-LEVEL SPATIAL SEMANTIC COMPLETION

To obtain a particle-level initial representation, we first employ Coord Net to map the $\mu_i^{*,t} \in G_t^*$ into coordinate features $X_{\text{Co}}^t \in \mathbb{R}^{N^* \times H_{\text{Co}}}$, which serve as the initial embedding for the Key Points:

$$X_{\text{Co}}^t = \text{ReLU}\left(G_t^* \mathbf{W}_1^\top + \mathbf{b}_1\right) \cdot \mathbf{W}_2^\top + \mathbf{b}_2 \tag{1}$$

To enhance DyG$^2$T's awareness of Key Points' spatial distribution for accurate geometry appearance prediction, we adopt a PosDiff Encoder to learn pairwise relative distances as spatial biases to enrich the initial Key Point embeddings. Specifically, we encode the coordinate differences between Key Points using a 2-layer MLP with ReLU. The pairwise encodings are reduced to nodewise via neighbor-wise mean aggregation and added to the coordinate features $X_{\text{Co}}^t$, yielding the enhanced coordinate feature $X_{\text{CoP}}^t \in \mathbb{R}^{N^* \times H_{\text{Co}}}$. $H$ denotes the feature embedding dimensions, and the subscript Co is introduced for distinction.

Relying on the coordinate features of sparse Key Points will lose the fine-grained edge features encoded by the Raw Point Cloud. Therefore, as shown in Figure 2(b), we utilize PointNet Qi et al. (2017a) to aggregate the $k$-nearest neighbors of each Key Point from the Raw Point Cloud, yielding neighborhood features $X_{\text{Po}}^t \in \mathbb{R}^{N^* \times H_{\text{Po}}}$. Specifically, for each Key Point, we apply a shared MLP to encode the coordinates of its $k$-nearest neighboring Raw Points. A feature transformation module is then introduced to map these coordinate features from different frames into a unified neighborhood feature space. Finally, a max-pooling operation is performed over the $k$ neighborhood features of each Key Point $i$ at frame $t$, yielding the representation $X_{\text{Po},i}^t \in \mathbb{R}^{H_{\text{Po}}}$.

Furthermore, to address the inconsistency of feature spaces between coordinate features $X_{\text{CoP}}^t$ and neighborhood features $X_{\text{Po}}^t$, we design a Multi-head Position-Aware Attention mechanism that en-

ables cross-space feature fusion guided by Gaussian-weighted relations $\omega$ Yang et al. (2025). For clarity, we illustrate the process using a single-head example:

$$Q = X_{\text{CoP}}^t \mathbf{W}^Q, K = X_{\text{CoP}}^t \mathbf{W}^K, V = X_{\text{Po}}^t \mathbf{W}^V \tag{2}$$

$$X_{\text{In}}^t = \text{SoftMax}\left(\frac{Q \cdot K^{\mathsf{T}}}{\sqrt{H_K}} + \log(\omega)\right) V \tag{3}$$

where $\mathbf{W}^Q, \mathbf{W}^K \in \mathbb{R}^{H_{\text{Co}} \times H_K}$ and $\mathbf{W}^V \in \mathbb{R}^{H_{\text{Po}} \times H_V}$ are learnable matrices, $\omega = \exp\left(-\frac{d^2}{2\rho^2}\right)$, $d$ is the Euclidean distance between Key Points and $\rho$ is a sharpness coefficient. This mechanism allows neighborhood features to be aggregated in a spatially-aware manner guided by the prior $\omega$. $X_{\text{In}}^t \in \mathbb{R}^{N^* \times H_{\text{In}}}$ is the spatial semantic features at frame $t$. Applying the same procedure to Key Points at $t-2$ and $t-1$, we obtain the spatial semantic feature sequence $\{X_{\text{In}}^{t-2}, X_{\text{In}}^{t-1}, X_{\text{In}}^t\}$.

### 3.2.2 OBJECT-LEVEL DYNAMIC TEMPORAL AGGREGATION

Considering that forces such as collisions can cause object deformations and alter motion trajectories, we employ Motion Align Net (MAN) to compute the relative motion supplement. Specifically, we take frame $t-1$ as the reference and compute the relative motion offsets $\delta^{t-2}, \delta^t \in \mathbb{R}^{N^* \times H_{\text{In}}}$ for frames $t-2$ and $t$. These offsets are added to the spatial semantic features $X_{\text{In}}^{t-2}$ and $X_{\text{In}}^t$, respectively, to compensate for the motion space gaps caused by motion across frames:

$$X_{\text{InA}}^{t-2} = X_{\text{In}}^{t-2} + \delta^{t-2}, \ X_{\text{InA}}^t = X_{\text{In}}^t + \delta^t \tag{4}$$

$$\delta^{t-2}, \delta^t = \tanh\left(\text{MAN}\left(X_{\text{InC}}; \ \mathbf{W}_{\text{In}}, \mathbf{b}_{\text{In}}\right)\right) \tag{5}$$

where $X_{\text{InC}} = \text{Concat}\left(X_{\text{In}}^{t-2}, X_{\text{In}}^{t-1}, X_{\text{In}}^t\right) \in \mathbb{R}^{N^* \times 3H_{\text{In}}}$ is cross-frame feature. The learnable weight $\mathbf{W} \in \mathbb{R}^{2H_{\text{In}} \times 3H_{\text{In}}}$ and bias $\mathbf{b} \in \mathbb{R}^{2H_{\text{In}}}$ are employed in the mapping process. The $\tanh(\cdot)$ constrains the magnitude of motion offset to prevent excessive correction. The function MAN projects the cross-frame feature $X_{\text{InC}}$ into a unified feature space and partitions it into blocks, from which relative motion offsets $\delta^{t-2}$ and $\delta^t$ are derived under the $\tanh(\cdot)$ constraint. $X_{\text{InA}}^{t-2}$ and $X_{\text{InA}}^t$ represent the aligned features of frames $t-2$ and $t$. Zero correction is applied to $X_{\text{InA}}^{t-1} = X_{\text{In}}^{t-1}$, which serves as the alignment reference.

Subsequently, we apply Temporal Attention to aggregate the Key Points' features across frames:

$$X_{\text{Ag}} = \sum_{i=t-2}^t \frac{\exp(s_i)}{\sum_{j=t-2}^t \exp(s_j)} X_{\text{InA}}^i \tag{6}$$

$$s_i = \mathbf{W}_{n2}^{\mathsf{T}} \cdot \tanh\left(\mathbf{W}_{n1} X_{\text{InA}}^i + \mathbf{b}_{n1}\right) + \mathbf{b}_{n2} \tag{7}$$

where $X_{\text{Ag}} \in \mathbb{R}^{N^* \times H_{\text{Ag}}}$ represents the Key Point features after dynamic temporal aggregation.

### 3.3 DYNAMICS MODELING BASED ON PARTICLE GRAPH TRANSFORMER

In this section, we describe how the Particle Graph and $X_{\text{Ag}}$ are utilized to capture multi-scale force propagation patterns at a global scale, and to predict the translation vectors $M^{*,t} \in \mathbb{R}^{N^* \times 3}$ of Key Points at frame $t$, estimating the Key Point position $\hat{G}_{t+1}^*$ at frame $t+1$.

As shown in Figure 2(c), inspired by Shi et al. (2021), we introduce a Particle Graph Transformer. For the Particle Graph constructed from the Key Points, we add edges between each Key Point and its top-$k_G$ nearest neighbors within a distance threshold $d_e$. The presence or absence of edges using binary values $0, 1$. After vectorizing the binarized adjacency matrix, $e_{ij}$ is defined as the learnable embedding of the edge features between nodes $i$ and $j$. Then, we capture global force propagation through Graph Attention. Specifically, the aggregated Key Point features $X_{\text{Ag}}$ are projected into the key $k^{(0)} \in \mathbb{R}^{N^* \times d_k}$, query $q^{(0)} \in \mathbb{R}^{N^* \times d_q}$, and value $v^{(0)} \in \mathbb{R}^{N^* \times d_v}$ using separate linear layers. The attention scores $\alpha$ are computed using a scaled dot-product function:

$$\alpha_{ij}^{(l)} = \frac{\langle q_i^{(l)}, k_j^{(l)} + e_{ij}\rangle}{\sum_{u \in \mathcal{N}(i)}\langle q_i^{(l)}, k_u^{(l)} + e_{iu}\rangle} \tag{8}$$

where $\langle q, k \rangle = \exp\left(\frac{q^\top k}{\sqrt{d_k}}\right)$, $i$ and $j$ are the endpoints of an edge, $i \in N^*$, $\mathcal{N}$ represents the set of neighbors of node $i$, and $l = 1, 2, \ldots, L$ denotes the index of the Particle Graph Transformer layer. Finally, we selectively aggregate node features over the entire graph to obtain $\hat{X}^{(l)} \in \mathbb{R}^{N^* \times H_G}$:

$$\hat{X}^{(l)} = \sum_{i \in N^*} \sum_{j \in \mathcal{N}(i)} \alpha_{i,j}^{(l)}(v_j^{(l)} + e_{i,j}) \tag{9}$$

The Particle Graph Transformer constructs direct multi-scale force propagation paths by performing attention interactions globally. This allows distinctive information to be preserved in particle representations, which is critical for decoding accurate motion trajectories. Furthermore, to mitigate feature homogenization in multi-scale force propagation, we incorporate Gated Residual between Particle Graph Transformer layers. Refer to the Appendix for details.

The encoded features $X^{(L)}$ from the Particle Graph Transformer are finally decoded by the Motion Predictor into Key Point displacement vectors $M^{*,t} \in \mathbb{R}^{N^* \times 3}$. Based on these vectors, we predict Key Point positions at frame $t + 1$ as $\hat{G}_{t+1}^* = \left\{ \hat{\mu}_i^{*,t+1} \mid 1 \leq i \leq N^* \right\}$, $\hat{\mu}_i^{*,t+1} = \mu_i^{*,t} + M_i^{*,t}$.

**Loss Function.** DyG$^2$T is optimized by minimizing the MSE losses $\|\cdot\|^2$ between the predicted Key Point positions $\hat{G}_{t+i}^*$ and the ground truth $G_{t+i}^*$ over the next $\epsilon$ frames:

$$\mathcal{L}_{\text{pred}} = \sum_{i=1}^{\epsilon} \left\| \hat{G}_{t+i}^* - G_{t+i}^* \right\|^2 \tag{10}$$

## 4 EXPERIMENT

### 4.1 EXPERIMENT SETTINGS

**Dataset.** We introduce Spring-Gaus dataset Zhong et al. (2024) and our Unity3D-Heterogeneous (Unity3D-H) dataset. The Spring-Gaus synthetic dataset contains multiple elastic objects with diverse appearances and materials, recorded as 30-frame $512 \times 512$ motion videos from 10 views, and provides 3D motion trajectory ground truth. The Spring-Gaus real-world part includes videos of five dolls, recorded as 20-frame $1920 \times 1080$ motion videos from 3 views. We also construct the Unity3D-H dataset using the simulation software Unity3D Wang et al. (2010), consisting of a polyhedron made in two materials, rendered as 30-frame $2098 \times 1327$ videos from 10 views. Following Spring-Gaus, the first 20 frames (visible during training) are used for dynamic reconstruction (10 frames for the Spring-Gaus real-world dataset), while the unseen final 10 frames are reserved for evaluating dynamic prediction.

Table 1: Quantitative results of dynamic reconstruction for DyG$^2$T and baselines on Spring-Gaus synthetic dataset.

| | CD↓ | | EMD↓ | |
|---|---|---|---|---|
| Objects | Spring-Gaus | DyG$^2$T(Ours) | Spring-Gaus | DyG$^2$T(Ours) |
| Torus | 0.012 | **0.008** | 0.003 | **0.001** |
| Cross | 0.016 | **0.010** | 0.005 | **0.002** |
| Cream | 0.014 | **0.012** | 0.007 | **0.005** |
| Apple | 0.014 | **0.011** | 0.006 | **0.003** |
| Paste | 0.011 | **0.008** | 0.003 | **0.002** |
| Chess | 0.017 | **0.010** | 0.007 | **0.002** |
| Banana | 0.049 | **0.007** | 0.027 | **0.002** |
| Mean | 0.019 | **0.010** | 0.008 | **0.003** |

**Metrics.** Following prior work Zhang et al. (2025a), we adopt CD, computed the bidirectional $L2$ distance between the predicted and ground-truth point clouds, and EMD, which quantifies the minimal transformation cost between the two point clouds, as metrics for 3D trajectory evaluation. For 2D appearance evaluation, we use PSNR, SSIM Wang et al. (2004), and LPIPS Zhang et al. (2018) to assess the similarity between reasoning and ground-truth images from different points of view. Since the baselines do not provide evaluation code, we reimplement and evaluate all methods within a unified framework to ensure fair comparison. Specificity, 3D metrics are averaged over all evaluated frames, while 2D metrics are first averaged across views per frame and then across frames.

### 4.2 DYNAMIC RECONSTRUCTION OF MOVING OBJECTS

To evaluate the dynamic reconstruction of DyG$^2$T, we primarily use CD and EMD to assess the quality of the reconstructed 3D trajectories. Quantitative results are presented in Table 1. The results demonstrate DyG$^2$T's ability to reconstruct 3D trajectories accurately across objects with varying appearances. Refer to the Appendix for more results.

Table 2: Quantitative results of motion prediction on Spring-Gaus synthetic dataset.

| Metrics | Methods | Torus | Cross | Cream | Apple | Paste | Chess | Banana | Mean |
|---|---|---|---|---|---|---|---|---|---|
| CD↓ | Spring-Gaus | 0.033 | 0.046 | 0.032 | 0.047 | 0.068 | 0.053 | 0.184 | 0.066 |
| | GS-Dynamics | 0.073 | 0.122 | 0.154 | 0.043 | 0.227 | 0.284 | 0.328 | 0.176 |
| | DyG$^2$T(Ours) | **0.029** | **0.038** | **0.027** | **0.028** | **0.038** | **0.050** | **0.055** | **0.039** |
| EMD↓ | Spring-Gaus | 0.014 | 0.024 | 0.023 | 0.029 | 0.035 | 0.027 | 0.091 | 0.035 |
| | GS-Dynamics | 0.033 | 0.062 | 0.097 | 0.021 | 0.164 | 0.200 | 0.171 | 0.107 |
| | DyG$^2$T(Ours) | **0.013** | **0.018** | **0.013** | **0.015** | **0.020** | **0.021** | **0.029** | **0.019** |
| PSNR↑ | Spring-Gaus | 12.220 | **11.993** | 11.267 | 17.443 | 11.016 | 11.305 | 15.949 | 13.028 |
| | GS-Dynamics | 13.450 | 10.621 | 12.647 | 19.632 | 11.506 | 11.758 | 16.622 | 13.748 |
| | DyG$^2$T(Ours) | **14.048** | 11.632 | **14.765** | **20.477** | **14.698** | **15.653** | **17.904** | **15.587** |
| SSIM↑ | Spring-Gaus | 0.850 | **0.876** | 0.709 | 0.828 | 0.775 | 0.755 | 0.865 | 0.808 |
| | GS-Dynamics | 0.876 | 0.842 | 0.763 | 0.887 | 0.802 | 0.749 | 0.880 | 0.828 |
| | DyG$^2$T(Ours) | **0.895** | 0.871 | **0.875** | **0.907** | **0.887** | **0.873** | **0.919** | **0.889** |
| LPIPS↓ | Spring-Gaus | 0.349 | 0.303 | 0.370 | 0.230 | 0.332 | 0.335 | 0.250 | 0.310 |
| | GS-Dynamics | 0.197 | 0.280 | 0.324 | 0.163 | 0.317 | 0.306 | 0.210 | 0.257 |
| | DyG$^2$T(Ours) | **0.139** | **0.220** | **0.189** | **0.131** | **0.178** | **0.207** | **0.122** | **0.171** |

Table 3: Quantitative results of motion prediction on Spring-Gaus real-world dataset and our Unity3D-H dataset.

| Metrics | Methods | Spring-Gaus real-world | | | | | Unity3D-H |
|---|---|---|---|---|---|---|---|
| | | Dog | Potato | Pig | Burger | Bun | Polyhedron |
| PSNR↑ | Spring-Gaus | 21.499 | 20.881 | 21.136 | 21.026 | 20.456 | 31.027 |
| | GS-Dynamics | 26.141 | **28.623** | 27.114 | 27.969 | 26.929 | 31.352 |
| | DyG$^2$T(Ours) | **27.676** | 27.933 | **27.750** | **30.645** | **27.197** | **31.768** |
| SSIM↑ | Spring-Gaus | 0.987 | 0.985 | 0.986 | 0.985 | 0.984 | 0.985 |
| | GS-Dynamics | 0.988 | **0.989** | 0.989 | 0.988 | 0.988 | 0.987 |
| | DyG$^2$T(Ours) | **0.991** | 0.987 | **0.989** | **0.994** | **0.988** | **0.990** |
| LPIPS↓ | Spring-Gaus | 0.030 | 0.032 | 0.031 | 0.031 | 0.032 | 0.020 |
| | GS-Dynamics | 0.023 | **0.020** | 0.020 | 0.022 | 0.020 | 0.018 |
| | DyG$^2$T(Ours) | **0.019** | 0.021 | **0.017** | **0.012** | **0.018** | **0.015** |

## 4.3 DYNAMICS MODELING AND REASONING OF MOVING OBJECTS

To evaluate dynamic reasoning, we conduct motion predictions on both Spring-Gaus and Unity3D-H datasets. As shown in Table 2, DyG$^2$T achieves promising results on the Spring-Gaus synthetic dataset, particularly in 3D trajectory reasoning (CD & EMD), highlighting the enhanced trajectory prediction accuracy through more faithful dynamics modeling. Although DyG$^2$T ranks second on Cross in terms of PSNR and SSIM, it still outperforms all methods on LPIPS, which better

Table 4: Quantitative results of motion prediction on the Heterogeneous Torus in Spring-Gaus synthetic dataset. DyG$^2$T$_{noisy}$ uses noisy dynamic reconstruction results to evaluate sensitivity to noise input.

| Methods | CD↓ | EMD↓ | PSNR↑ | SSIM↑ | LPIPS↓ |
|---|---|---|---|---|---|
| Spring-Gaus | 0.030 | 0.012 | 12.197 | 0.850 | 0.350 |
| GS-Dynamics | 0.116 | 0.049 | 13.342 | 0.861 | 0.235 |
| DyG$^2$T$_{noisy}$ | 0.035 | 0.016 | 13.647 | 0.878 | 0.188 |
| DyG$^2$T(Ours) | **0.015** | **0.000** | **14.080** | **0.893** | **0.129** |

aligns with human perceptual similarity. Figure 3(a) further illustrates that the predictions of Spring-Gaus (spring-mass model) Zhong et al. (2024) and GS-Dynamics (GNN-based simulator) Zhang et al. (2025a) exhibit noticeable positional deviations, whereas DyG$^2$T accurately infers trajectories and fine-grained appearance details, demonstrating strong generalizability across diverse objects. Moreover, the inference time (Time) and frame rate (FPS) in Figure 3(a) also confirm that DyG$^2$T performs dynamic reasoning with higher computational efficiency.

Furthermore, as shown in Table 3, the evaluation on the Spring-Gaus real-world and Unity3D-H datasets demonstrates that DyG$^2$T exhibits strong real-world generalization and can be readily transferred to other benchmarks. While GS-Dynamics achieves leading performance on the relatively simple Potato, it falls behind on toys with more complex geometries, such as Burger.

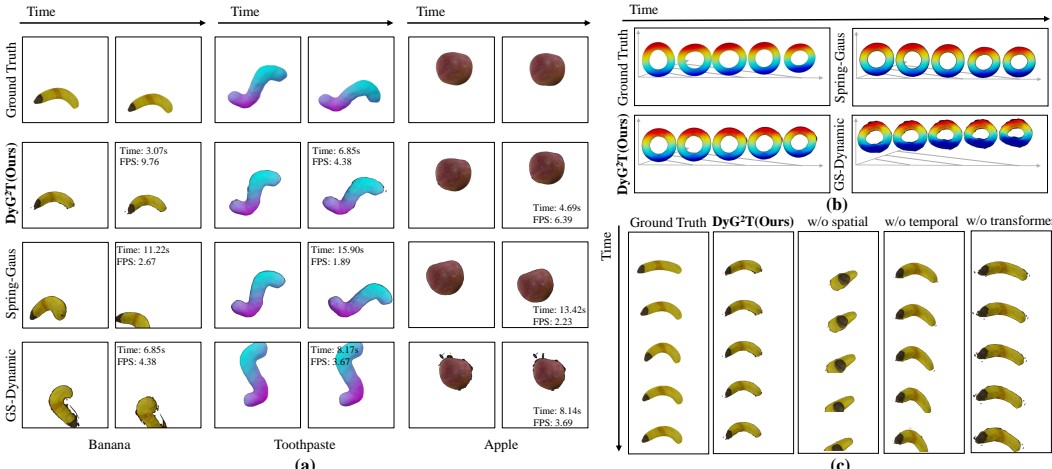

Figure 3: Qualitative results. Motion prediction of (a) homogeneous, (b) heterogeneous objects, and (c) DyG$^2$T ablation variant in the Spring-Gaus synthetic dataset.

To further evaluate robustness and generalization, we conduct experiments on a Heterogeneous Torus with complex physical properties. As shown in Table 4, DyG$^2$T achieves more accurate trajectory prediction than the methods by capturing the intricate force propagation within heterogeneous objects, yielding trajectories closer to the ground truth (Figure 3(b)).

Table 5: Quantitative results of module ablation study.

| Methods | CD↓ | EMD↓ | PSNR↑ | SSIM↑ | LPIPS↓ |
|---|---|---|---|---|---|
| w/o spatial | 0.354 | 0.203 | 16.174 | 0.875 | 0.220 |
| w/o temporal | 0.227 | 0.120 | 16.493 | 0.882 | 0.198 |
| w/o transformer | 0.197 | 0.110 | 16.246 | 0.875 | 0.201 |
| DyG$^2$T(Ours) | **0.055** | **0.029** | **17.904** | **0.919** | **0.122** |

Moreover, we assess sensitivity to noisy inputs by introducing perturbations to the central trajectory and applying non-rigid deformations to the dynamic reconstruction results. While noise causes only minor degradation in 3D trajectory prediction for DyG$^2$T$_{\text{noisy}}$, its 2D appearance evaluation still surpasses the baselines. The results on the Unity3D-H dataset in Table 3 also validate DyG$^2$T's scalability to heterogeneous objects across different benchmarks.

### 4.4 ABLATION STUDIES

**The Modules of DyG$^2$T.** The ablation results for DyG$^2$T's core modules are presented in Table 5 and Figure 3(c). Compared to DyG$^2$T, the w/o spatial variant, where the neighborhood feature $X_{\text{Po}}$ is disabled, exhibits a significant performance drop. This confirms that sparse sampling from the Raw Point cloud causes a notable loss of fine-grained edge information and spatial semantics, and the Particle-level Spatial Semantic Completion mechanism effectively addresses this issue. Similarly, the performance degradation of w/o temporal, which replaces the temporal attention aggregation with simple feature concatenation, highlights the

Table 6: Ablation study on neighborhood ranges $k$ (Row 2 & 3), the Particle-level Spatial Semantic Completion (Row 4 & 5), and the Object-level Dynamic Temporal Aggregation (Row 6~9).

| Methods | CD↓ | EMD↓ | PSNR↑ | SSIM↑ | LPIPS↓ |
|---|---|---|---|---|---|
| DyG$^2$T$_{k=8}$ | 0.244 | 0.125 | 16.368 | 0.878 | 0.207 |
| DyG$^2$T$_{k=32}$ | 0.169 | 0.092 | 16.365 | 0.879 | 0.190 |
| w/o PosDiff | 0.372 | 0.178 | 16.742 | 0.886 | 0.195 |
| w/o PosAware | 0.107 | 0.056 | 17.613 | 0.912 | 0.139 |
| w/o MAN | 0.107 | 0.240 | 16.071 | 0.871 | 0.242 |
| LSTM | 0.117 | 0.055 | 17.002 | 0.893 | 0.166 |
| avg pool | 0.096 | 0.052 | 17.344 | 0.906 | 0.145 |
| max pool | 0.365 | 0.195 | 16.290 | 0.874 | 0.221 |
| DyG$^2$T(Ours) | **0.055** | **0.029** | **17.904** | **0.919** | **0.122** |

effectiveness of the Object-level Dynamic Temporal Aggregation in preserving dynamic evolution patterns. Moreover, the comparison between DyG$^2$T and w/o transformer, which replaces the Particle Graph Transformer with a vanilla GNN, demonstrates the importance of modeling force propagation across multi-scales. Due to GNN's limitations in local modeling, w/o transformer fails to perform effective dynamics modeling, even when enriched with spatiotemporal features.

**Spatial-Temporal Feature Completion and Aggregation.** We investigate how the range $k$ of Raw Point neighborhood features $X_{\text{Po}}^t$ influences dynamics modeling. As shown in Table 6 row 2 &

3, we observe that an insufficient neighborhood ($DyG^2T_{k=8}$) leads to inadequate spatial semantic completion, resulting in a more significant performance drop compared to the large neighborhood ($DyG^2T_{k=32}$), which introduces redundant information. Thus, we set $k = 16$, which empirically serves as the best practice for $DyG^2T$.

We further conduct an ablation study on different components of the Particle-level Spatial Semantic Completion. As shown in Table 6 row 4 & 5, the w/o PosDiff variant, which disables the PosDiff Encoder, exhibits the most substantial degradation in CD and EMD. This highlights the importance of encoding relative positional information among Key Points. Furthermore, the comparison between w/o PosAware and $DyG^2T$ demonstrates that the Multi-head Position-Aware Attention mechanism effectively guides the integration of coordinate and neighborhood features via Gaussian-weighted $\omega$, enhancing the accuracy of dynamics modeling.

We also ablate the Motion Align Net (MAN) to investigate whether it successfully compensates for relative motion offset. As shown in Table 6 row 6~9, the w/o MAN variant fails to model object dynamics from misaligned motion features. This validates the effectiveness of MAN in compensating for potential inter-frame inconsistencies. Additionally, we compare $DyG^2T$'s Temporal Attention mechanism

Table 7: Ablation study of the number of Key Points $N^*$ in the Particle Graph.

| Variants | CD↓ | EMD↓ | PSNR↑ | SSIM↑ | LPIPS↓ |
|---|---|---|---|---|---|
| $N^* = 50$ | 0.470 | 0.247 | 16.003 | 0.869 | 0.248 |
| $N^* = 100$ | **0.055** | **0.029** | **17.904** | **0.919** | **0.122** |
| $N^* = 150$ | 0.098 | 0.050 | 17.509 | 0.909 | 0.151 |

with alternative feature aggregation strategies, including LSTM and average&max pooling. The results demonstrate that Temporal Attention achieves superior performance in aggregate temporally coherent motion patterns, whereas the alternatives suffer from feature ambiguity or loss.

**Dynamics Modeling Based on Particle Graph Transformer.** Table 7 presents the impact of the number of Key Points $N^*$ on dynamics modeling. A small Graph ($N^* = 50$) including sparse Key Points, which fail to provide sufficient support for dynamics modeling. Moreover, a large Graph ($N^* = 150$) introduces excessive redundancy, which can hinder effective modeling of dynamics. Table 8 investigates the effect of different edge sparsity

Table 8: Ablation study of the presence of edges between Key Points and their top-$k_G$ nearest neighbors.

| Variants | CD↓ | EMD↓ | PSNR↑ | SSIM↑ | LPIPS↓ |
|---|---|---|---|---|---|
| $k_G = 2$ | 0.564 | 0.255 | **18.436** | 0.910 | 0.183 |
| $k_G = 5$ | **0.055** | **0.029** | 17.904 | **0.919** | **0.122** |
| $k_G = 10$ | 0.186 | 0.107 | 16.398 | 0.881 | 0.193 |

levels in the Particle Graph. A sparse Particle Graph ($k_G = 2$) lacks sufficient alternative paths for modeling force propagation, while a dense graph ($k_G = 10$) increases the difficulty of identifying optimal propagation. Notably, the $k_G = 2$ variant exhibits poor trajectory prediction, causing the rendering images to be filled with a large number of invalid pixels and resulting in abnormally high PSNR; the LPIPS and SSIM still demonstrate the superiority of our method. Accordingly, we adopt a moderate value of $N^* = 100$, $k_G = 5$ as the best practice for $DyG^2T$.

## 5 CONCLUSION

This paper proposes $DyG^2T$, a dynamics modeling framework that integrates spatiotemporally completed particle representations with multi-scale force propagation modeling. Spatially, $DyG^2T$ enriches each Key Point feature with fine-grained edge information and geometry perception by aggregating positions from corresponding raw particles and relative coordinates from neighboring Key Points. Temporally, inter-frame relative motion offsets are computed via Motion Align Net, and Temporal Attention aggregates particle features across frames to preserve dynamic evolution patterns. A Particle Graph Transformer further captures long-range interactions through multi-scale force propagation paths, enabling accurate modeling of complex object dynamics. Extensive experiments on both synthetic and real-world datasets demonstrate that $DyG^2T$ achieves precise trajectory prediction while maintaining strong cross-object and real-world generalization. For future work, we plan to investigate adaptive optimization mechanisms for hyperparameters (*e.g.*, the neighborhood range $k$ of Raw Points), reducing reliance on prior knowledge. In addition, we aim to extend $DyG^2T$ to model the dynamics of more complex heterogeneous objects (*e.g.*, solid–liquid–gas mixtures), further enhancing its scalability to real-world scenarios.

REPRODUCIBILITY STATEMENT

To ensure the reproducibility of our work, we provide general details on the datasets and experimental settings in Section 4.1. Comprehensive information on the model architecture, datasets, baselines, training setup, and additional results can be found in the Appendix A.

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

# APPENDIX

## THE USAGE OF LLMS

In this work, large language models (LLMs) are employed solely during the manuscript preparation stage to assist with translation and language refinement. Beyond this purpose, they are not utilized for any other aspects of the study.

## A  OVERVIEW

We provide Implementation Details and Additional Results of DyG$^2$T in the Appendix. Specifically, the Appendix includes the following sections:

- Implementation details of the Trackable Particle-based Representation by Dynamic Reconstruction module, including:
    - A comprehensive introduction to 3D Gaussian Splatting, which serves as the theoretical foundation;
    - The initialization strategy for the first frame;
    - The physics-based prior loss functions used to constrain dynamic reconstruction;
    - Training settings used during dynamic reconstruction.
- Implementation details of the Spatial-Temporal Features Completion and Aggregation module, including:
    - Normalization procedures to ensure the stability of Key/Raw Points;
    - Details on PointNet-based encoding of $k$-nearest neighbors for each Key Point.
- Implementation details of the Dynamics Modeling based on the Particle Graph Transformer module, including:
    - The construction process of the Particle Graph;
    - The implementation of the Particle Graph Transformer;
    - Details on how Linear Blend Skinning (LBS) is used to predict dense motion point clouds.
- Datasets and baselines used to evaluate the performance of DyG$^2$T.
- Hyperparameter settings used during the training of DyG$^2$T.
- Configuration details of the released DyG$^2$T source code.
- Additional results of dynamic reconstruction and ablation studies of the physics-based prior constraints.
- An investigation on how different reference frames in Motion Align Net affect the dynamics modeling performance.
- Evaluation of DyG$^2$T and GNN-based baselines on dynamics modeling of heterogeneous materials.

# B IMPLEMENTATION DETAILS

## B.1 TRACKABLE PARTICLE-BASED REPRESENTATIONS BY DYNAMIC RECONSTRUCTION

### B.1.1 PRELIMINARY: 3D GAUSSIAN SPALTTING

3D Gaussian Splatting (3DGS) Kerbl et al. (2023) optimizes a set of learnable Gaussian kernels as an explicit scene representation through a differentiable rasterizer. Each kernel is parameterized by a center $\mu_i$, a covariance matrix $\Sigma_i$, an opacity $\sigma_i$, and spherical harmonics coefficients $\mathbf{h}_i$. The color $\mathbf{C}$ of each 2D pixel is computed using a depth-sorted Max Volume Rendering Luiten et al. (2024):

$$\mathbf{C} = \sum_{i \in N} \mathbf{c}_i \phi_i^{2D} \prod_{j=1}^{i-1} (1 - \phi_j^{2D}) \tag{11}$$

where $N$ denotes the set of the Gaussian kernels, and $\mathbf{c}_i$ is the RGB color of kernel $i$ obtained from spherical harmonics based on the viewing direction and coefficients $\mathbf{h}_i$. $\phi_i$ is the weighted opacity of kernel $i$:

$$\phi_i = \sigma_i \exp\left(-\frac{1}{2}(\mathbf{x} - \mu_i)^\mathsf{T} \Sigma_i^{-1} (\mathbf{x} - \mu_i)\right) \tag{12}$$

$\phi_i^{2D}$ is the 2D version of Equation 12. The pixel position is obtained by approximating a perspective projection of the 3D Gaussian's center $\mu_i$ and covariance matrix $\Sigma_i$:

$$\begin{aligned}
\mu_i^{2D} &= (K((W\mu_i)/(W\mu_i)_z))_{1:2} \\
\Sigma_i^{2D} &= (JW\Sigma_i W^T J^T)_{1:2,1:2}
\end{aligned} \tag{13}$$

where $W$ and $K$ represent the extrinsic and intrinsic parameters of the view camera, respectively, and $J$ is the Jacobian matrix of the projection transformation.

### B.1.2 FIRST FRAME POINT CLOUD INITIALIZATION STRATEGY

Most existing methods utilize COLMAP Schonberger & Frahm (2016) to obtain a coarse 3D estimation of the object as the initialization for 3D Gaussian Splatting. However, prior studies Zhong et al. (2024) have shown that such initialization typically leads to Gaussian kernels being predominantly distributed on the object's surface, primarily encoding appearance information. This surface-biased distribution hinders subsequent modeling of internal force propagation within the object. To address this issue, we randomly initialize 100,000 Gaussian kernels with random RGB colors within the 3D space corresponding to the object's initial position. This strategy ensures a more uniform spatial distribution of kernels, providing a better foundation for capturing internal dynamics.

### B.1.3 PHYSICAL CONSTRAINTS FOR DYNAMIC RECONSTRUCTION

To ensure physical consistency while fitting the appearance of dynamic objects across frames, we incorporate both appearance loss and physical constraints following a similar strategy to Dynamic 3D Gaussian (Dyn3DGS) Luiten et al. (2024). Specifically, we employ a weighted combination of L1 loss and Structural Similarity Index (SSIM) Wang et al. (2004) loss as the optimization objective for static appearance reconstruction on the first frame:

$$\mathcal{L}_{\text{vis}} = (1 - \lambda_{\text{vis}}) \mathcal{L}_1 + \lambda_{\text{vis}} \mathcal{L}_{\text{SSIM}} \tag{14}$$

where $\mathcal{L}_{\text{vis}}$ is simultaneously applied to constrain both the 2D rendered images and the semantic segmentation maps, $\lambda_{\text{vis}} = 0.2$. Moreover, to ensure that the Gaussians with frozen visual attributes can accurately capture the motion dynamics of the object, we introduce non-rigid physical modeling constraints inspired by Dyn3DGS. These include the local rigidity loss $\mathcal{L}_{\text{rigid}}$, local rotation similarity loss $\mathcal{L}_{\text{rot}}$, and local isometric loss $\mathcal{L}_{\text{iso}}$. As a key constraint to prevent arbitrary movement of the Gaussian kernels, the local rigidity loss $\mathcal{L}_{\text{rigid}}$ enforces each Gaussian and its spatial neighbors to undergo consistent rigid transformations. This ensures compatibility of Gaussian rotation and translation across frames:

$$\mathcal{L}_{i,j}^{\text{rigid}} = w_{i,j} \left\| (\mu_{j,t-1} - \mu_{i,t-1}) - R_{i,t-1} R_{i,t}^{-1} (\mu_{j,t} - \mu_{i,t}) \right\|_2 \tag{15}$$

$$\mathcal{L}_{\text{rigid}} = \frac{1}{k|\mathcal{S}|} \sum_{i \in \mathcal{S}} \sum_{j \in \text{knn}_{i;k_{Ph}};} \mathcal{L}_{i,j}^{\text{rigid}} \tag{16}$$

where $S$ is the set of Gaussian functions, $\text{knn}_{i;k}$ denotes the $k_{Ph}$-nearest neighbors of Gaussian $i$, and $w_{i,j}$ represents the unnormalized isotropic Gaussian weight factor:

$$w_{i,j} = \exp\left(-\lambda_w \left\|\mu_{j,0} - \mu_{i,0}\right\|_2^2\right) \tag{17}$$

where $\lambda_w = 2000$. The weight $w_{i,j}$ is initialized using the positions of the Gaussian kernels in the first frame, $\mu_{i,0}$ and $\mu_{j,0}$, and kept fixed at all subsequent frames. In this way, Dyn3DGS enforces local rigidity while still allowing non-rigid transformations at the object scale. Although Dyn3DGS implicitly enforces consistent rotation among a Gaussian kernel and its neighbors through $\mathcal{L}_{\text{rigid}}$, introducing an explicit constraint on this objective can further improve the convergence behavior of the model:

$$\mathcal{L}_{\text{rot}} = \frac{1}{k|\mathcal{S}|} \sum_{i \in \mathcal{S}} \sum_{j \in \text{knn}_{i;k}} w_{i,j} \left\|\hat{q}_{j,t}\hat{q}_{j,t-1}^{-1} - \hat{q}_{i,t}\hat{q}_{i,t-1}^{-1}\right\|_2 \tag{18}$$

where $q$ denotes the rotation of a Gaussian kernel in quaternion form, and the Gaussian weight factor $w_{i,j}$ is shared with $\mathcal{L}_{\text{rigid}}$. Furthermore, to prevent dynamic point cloud tearing and separation that may arise from the continuous application of the local rigidity loss $\mathcal{L}_{\text{rigid}}$ and the local rotation similarity loss $\mathcal{L}_{\text{rot}}$ across adjacent frames, the local isometry loss $\mathcal{L}_{\text{iso}}$ is introduced. It enforces the consistency of relative distances among neighboring Gaussian kernels with respect to the first frame, thereby ensuring point cloud stability throughout long-term motion:

$$\mathcal{L}_{\text{iso}} = \frac{1}{k|\mathcal{S}|} \sum_{i \in \mathcal{S}} \sum_{j \in \text{knn}_{i;k}} w_{i,j} \left| \left\|\mu_{j,0} - \mu_{i,0}\right\|_2 - \left\|\mu_{j,t} - \mu_{i,t}\right\|_2 \right| \tag{19}$$

Overall, the dynamic reconstruction constraint is formulated as a weighted sum of all the aforementioned losses. For clarity, we use $\lambda_{\text{im}}$, $\lambda_{\text{seg}}$, $\lambda_{\text{rigid}}$, $\lambda_{\text{rot}}$, and $\lambda_{\text{iso}}$ to denote the weights corresponding to $\mathcal{L}_{\text{im}}$, $\mathcal{L}_{\text{seg}}$, $\mathcal{L}_{\text{rigid}}$, $\mathcal{L}_{\text{rot}}$, and $\mathcal{L}_{\text{iso}}$, respectively. Contrary to prior empirical practices, our experiments reveal that the 2D rendered images and semantic segmentation maps should be assigned equal weights; assigning inappropriate or excessively high weights can degrade the object appearance and introduce significant noise. In addition, the selection of $k_{Ph}$-nearest neighbors for Gaussian $i$ also has a notable impact on dynamic reconstruction performance.

As shown in Appendix B.1.3, we determine the optimal set of dynamic tracking parameters for the Spring-Gaus synthetic dataset through ablation studies. In practice, we set $\lambda_{\text{im}} = \lambda_{\text{seg}} = 1$, $\lambda_{\text{rigid}} = 4$, $\lambda_{\text{rot}} = 0.08$, $\lambda_{\text{iso}} = 20$, and $k_{Ph} = 20$.

### B.1.4 DYNAMIC RECONSTRUCTION CONFIGS

We set the number of iterations for the static reconstruction of the first frame to 10,000, during which all Gaussian kernel attributes are allowed to be optimized. Subsequently, except for the centers and rotations, all other attributes are frozen, and the iteration is reduced to 2,000. The dynamic reconstruction process consists of frame-by-frame optimization performed on the same set of Gaussian kernels. The centers and rotations for the next frame are estimated based on the motion vectors relative to the previous frame. The momentum parameters of the Adam optimizer are reinitialized at each frame.

### B.2 SPATIAL-TEMPORAL FEATURES COMPLETION AND AGGREGATION

### B.2.1 RAW POINT CLOUD CONSTRUCTION

To mitigate the impact of outliers on dynamics modeling, we introduce an outlier detection and filtering system based on the Median Absolute Deviation (MAD). Specifically, for a temporal sequence $\mathbf{P} = \{\mathbf{p}_0, \mathbf{p}_1, \cdots, \mathbf{p}_{T-1}\}$ where $\mathbf{p}_t \in \mathbb{R}^3$, we first quantify the relative motion $\mathbf{d} \in \mathbb{R}^{(T-1) \times N}$ of the point cloud using inter-frame Euclidean distances:

$$d_t = \left\|\mathbf{p}_{t+1} - \mathbf{p}_t\right\|_2 \quad \text{for} \quad t = 0, 1, \cdots, T-2 \tag{20}$$

The per-frame relative motion $d$ is accumulated along the temporal dimension to compute the total relative motion $s \in \mathbb{R}^N$ for each point:

$$s_n = \sum_{t=0}^{T-2} d_t^{(n)} \quad \text{for} \quad n = 0, 1, \cdots, N-1 \tag{21}$$

Subsequently, we apply a Median Absolute Deviation (MAD)-based outlier detection method to filter the point cloud:

$$
\begin{aligned}
\mu &= \text{median}(\mathbf{s}) \\
\text{MAD} &= \text{median}(|s_i - \mu|) \\
z_i &= |s_i - \mu|/\text{MAD} \\
\text{valid}_i &= \mathcal{I}[z_i < 3]
\end{aligned}
\tag{22}
$$

where $\mathcal{I}$ denotes that $\text{valid}_i$ is set to 1 when $z_i < 3$. In this way, we effectively filter out abnormal trajectory points that exhibit sudden jumps or remain stationary, ensuring the consistency of motion patterns across trajectory points.

Furthermore, we apply a 3-point moving average to suppress high-frequency noise and smooth local fluctuations in the trajectories:

$$\mathbf{P}_{1:T-1,:}^{(m)} = \frac{1}{3}\left(\mathbf{P}_{0:T-2,:}^{(m-1)} + \mathbf{P}_{1:T-1,:}^{(m-1)} + \mathbf{P}_{2:T,:}^{(m-1)}\right) \tag{23}$$

where $m = 1, \ldots, 10$ denotes the number of smoothing iterations. This method essentially functions as a low-pass filter for discrete signals, analogous to a convolution operation:

$$\mathbf{p}^{\text{new}} = \mathbf{p} * \mathbf{h} \tag{24}$$

where $\mathbf{h} = \left[\frac{1}{3}, \frac{1}{3}, \frac{1}{3}\right]$. The transfer function of this filter in the frequency domain is:

$$H(\omega) = \frac{1}{3}(1 + e^{-i\omega} + e^{-i2\omega}) \tag{25}$$

Through the above method, we extracted the Raw Point Cloud from the tracking reconstruction results of Dyn3DGS. Subsequently, we applied Farthest Point Sampling (FPS) Qi et al. (2017b) to downsample the Raw Point Cloud containing $N$ points into a set of Key Points with $N^*$ points. In addition, to avoid overly dense and imbalanced distributions of key points, we employed a distance-based filtering strategy to remove clustered points whose pairwise distances fall below a predefined threshold.

### B.2.2 $k$-NEAREST-NEIGHBOR SPATIAL SEMANTIC ENCODING

We adopt PointNet Qi et al. (2017a) as the local spatial-semantic encoder for the Key Point neighborhoods. To better tailor it to our task, we make slight modifications to the vanilla PointNet. Specifically, we extract the $N^* \times d_{\text{PN}}$ hidden features before the global pooling layer of the vanilla PointNet Classification Network as the spatial-semantic representations of the $N^*$ Key Points, where $d_{\text{PN}}$ is a hyperparameter of DyG$^2$T, typically set to 256 in practice.

Moreover, we disable the T-Net module (Input Transform), originally designed to predict affine transformation matrices in vanilla PointNet. This is because DyG$^2$T already ensures affine alignment and cross-frame translation consistency via the Motion Align Net. Introducing an additional T-Net would lead to redundant correction and may negatively affect motion coherence.

### B.3 DYNAMICS MODELING BASED ON PARTICLE GRAPH TRANSFORMER

### B.3.1 PARTICLE GRAPH CONSTRUCTION

We construct the Particle Graph by adding undirected edges between Key Points whose pairwise distances fall below a predefined threshold $d_e$. Specifically, we compute the Euclidean distances between all pairs of Key Points at frame $t$ and connect each point to the top-$k_G$ closest neighbors with distances less than $d_e$. In practice, we set $d_e = 0.08$ (with $d_e = 0.1$ for Cross), and the value of top-$k_G$ is determined based on the object category: for Cross, Apple, Toothpaste, and Chess, $k_G = 7$; for all other objects, $k_G = 5$. The size of the Particle Graph is denoted as $N^*$, with its optimal value varying across object appearances: $N^* = 100$ for Torus, Cream, and Banana; $N^* = 120$ for Cross; and $N^* = 150$ for Apple, Toothpaste, and Chess.

### B.3.2 Force Propagation Based on Particle Graph Transformer

As shown in Figure 2 of the main text, the aggregated output of the Spatial-Temporal Features Completion and Aggregation mechanism module, denoted as $X_{\text{Ag}} \in \mathbb{R}^{N^* \times H_{\text{Ag}}}$, serves as the node feature input for force propagation modeling. Before performing computations within the Particle Graph Transformer, we introduce a Particle Encoder to bridge the learnable feature gap between the two modules. This encoder is implemented as a 3-layer MLP with ReLU, which maps the feature space from $H_{\text{Ag}}$ to $H_G$.

Our Particle Graph Transformer for dynamics modeling is built upon the UniMP architecture proposed by Shi et al. Shi et al. (2021), with several key adaptations. Specifically, we disable the Masked Label Prediction mechanism and retain the core Particle Graph Transformer module for global interaction modeling, along with the Gated Residual to mitigate over-smoothing. Furthermore, recognizing the unordered nature of point cloud particles in dynamic modeling—*i.e.*, the invariance to the ordering of nodes in the adjacency or feature matrices—we also remove the Rotary Embedding strategy originally used for node positional encoding in UniMP. To mitigate the over-smoothing that often arises in full-graph modeling, we introduce Gated Residual Chen et al. (2020); Li et al. (2019a):

$$g^{(l)} = \mathbf{W}_g^{(l)} \hat{X}^{(l)} + \mathbf{b}_g^{(l)} \tag{26}$$

$$\beta^{(l)} = \text{Sigmod}\left(\mathbf{W}_\beta^{(l)} \left[\hat{X}^{(l)}; g^{(l)}; \hat{X}^{(l)} - g^{(l)}\right]\right) \tag{27}$$

$$X^{(l+1)} = \max\left(\text{LN}\left(\left(1 - \beta^{(l)}\right) \hat{X}^{(l+1)} + \beta^{(l)} g^{(l)}\right)\right) \tag{28}$$

### B.3.3 Dense 3D Gaussian Motion Prediction

We need to estimate the 3D rotation $R^{*,t} \in \mathbb{R}^{N^* \times 3}$ based on the key point translation motion $M^{*,t}$:

$$R_i^{*,t} = \arg \min_{R \in SO(3)} \sum_{j \in \mathcal{N}^*(i)} \left\| R\left(\mu_j^{*,t} - \mu_i^{*,t}\right) - \left(\mu_j^{*,t+1} - \mu_i^{*,t+1}\right) \right\|^2 \tag{29}$$

where $\mathcal{N}^*(i)$ is the local neighborhood of key point $i$. Next, we use Linear Blend Skinning (LBS) Sumner et al. (2007); Huang et al. (2024) to interpolate the densified Gaussian kernel at $t + 1$, $G_{t+1} = \left\{\mu_i^{t+1}\right\}_{0 \leq i \leq N}$, based on the 6-DoF transformation of key points at $t$ (*i.e.*, $M^{*,t}$ and $R_i^{*,t}$). Specifically:

$$\mu_i^{t+1} = \sum_{u=1}^{N^*} \gamma_{iu}^t \left(R_u^t \left(\mu_i^t - \mu_u^{*,t}\right) + \mu_u^{*,t} + M^{*,t}\right)$$

$$r_i^{t+1} = \left(\sum_{u=1}^{N^*} \gamma_{iu}^t f\left(R_u^{*,t}\right)\right) \odot r_i^t \tag{30}$$

where $f\left(R_u^{*,t}\right)$ denotes the mapping from the rotation matrix of key point $u$ at $t$ to its quaternion representation. $\mu_i^{t+1}$ and $r_i^{t+1}$ represent the center and rotation quaternion of the Gaussian kernel $i$ at $t + 1$, respectively. The weight $\gamma_{iu}^t = \frac{\left\|\mu_i^t - \mu_b^{*,t}\right\|^{-1}}{\sum_{u=1}^{N^*} \left\|\mu_i^t - \mu_u^{*,t}\right\|^{-1}}$ captures the relative influence of key point $u$ on Gaussian kernel $i$.

### B.4 Dataset & Baselines

**Dataset.** We evaluate DyG$^2$T on Spring-Gaus synthetic & real-world dataset Zhong et al. (2024) and Unity3D-Heterogeneous dataset. In the Spring-Gaus synthesis dataset, the initial 3D appearance is derived from PAC-NeRF Li et al. (2023) and OmniObject3D Wu et al. (2023). Then, MPM is used to simulate dynamics and obtain dynamic ground truth, followed by multi-view RGB video rendering using Blender. For the Unity3D-Heterogeneous dataset, we first constructed a polyhedron in Unity3D Wang et al. (2010) composed of heterogeneous elastic materials in a 1:1 ratio. Subsequently, we strategically placed ten synchronized cameras across the upper hemisphere of the scene to capture the motion of the polyhedron being released from midair and bouncing upon impact with the ground. Unlike the Spring-Gaus synthetic dataset, which simulates 3D point clouds

Table 9: Quantitative results of dynamic reconstruction for DyG$^2$T and baselines on Spring-Gaus synthetic dataset.

| Metrics | Methods | Torus | Cross | Cream | Apple | Paste | Chess | Banana | Mean |
|---|---|---|---|---|---|---|---|---|---|
| CD↓ | Dy-Gaus | 579 | 773 | 479 | 727 | 2849 | 764 | 2963 | 1305 |
| | 4D-Gaus | 11.12 | 1.77 | 2.87 | 2.23 | 1.95 | 3.97 | 7.13 | 4.43 |
| | PAC-NeRF | 4.92 | 1.10 | 0.77 | 1.11 | 3.14 | 0.96 | 2.77 | 2.11 |
| | Spring-Gaus | 0.012 | 0.016 | 0.014 | 0.014 | 0.110 | 0.017 | 0.049 | 0.019 |
| | DyG$^2$T(Ours) | **0.008** | **0.010** | **0.012** | **0.011** | **0.008** | **0.010** | **0.007** | **0.010** |
| EMD↓ | Dy-Gaus | 0.857 | 0.995 | 0.783 | 0.903 | 1.739 | 0.985 | 1.591 | 1.116 |
| | 4D-Gaus | 0.130 | 0.078 | 0.089 | 0.088 | 0.070 | 0.097 | 0.112 | 0.095 |
| | PAC-NeRF | 0.056 | 0.052 | 0.041 | 0.045 | 0.054 | 0.052 | 0.062 | 0.052 |
| | Spring-Gaus | 0.003 | 0.005 | 0.007 | 0.006 | 0.003 | 0.007 | 0.024 | 0.008 |
| | DyG$^2$T(Ours) | **0.001** | **0.002** | **0.005** | **0.003** | **0.002** | **0.002** | **0.002** | **0.003** |

before rendering videos, Unity3D directly simulates and renders the dynamics of the object based on its heterogeneous material properties. Consequently, the Unity3D-Heterogeneous dataset does not provide 3D point cloud data that could be used to evaluate trajectory consistency.

**Baselines.** Following previous works, we evaluate DyG$^2$T from two perspectives. For dynamic reconstruction, we compare against Spring-Gaus Zhong et al. (2024), which optimizes per-frame geometry using a spring-mass model initialized from the first frame. Referring to the Spring-Gaus Zhong et al. (2024), we also introduced Dy-Gaus Luiten et al. (2024), 4D-Gaus Wu et al. (2024), and PAC-NeRF Li et al. (2023) to more comprehensively evaluate the dynamic reconstruction performance. For dynamics modeling, we adopt GS-Dynamics Zhang et al. (2025a), a pipeline that combines Dyn3DGS with GNNs, and Spring-Gaus as baselines. All baselines are retrained using the optimal hyperparameters recommended by the authors, and the best performances are reported.

## B.5 TRAINING SETUP

We train DyG$^2$T for 1000 epochs, with each epoch comprising 100 iterations Zhang et al. (2025a). We adopt the Adam optimizer with a learning rate of 0.001. The PointNet uses $k = 16$ ($k = 8$ in Apple and Toothpaste), and the sharpness parameter $\rho$ is set to half the minimum distance between key points. We perform $\epsilon = 5$ predictions in each training iteration and sum the MSE losses, which are used for backpropagation. To meet training requirements, we augment each dynamic reconstructed object trajectory to 30 instances via random translation and rotation, and split them into training and validation sets at a 4:1 ratio.

## B.6 CODE IMPLEMENTATION

The source codes and corresponding operation instructions are shown in code.zip. The results of the submitted codes are achieved in the following environment:

$$
\begin{aligned}
\text{cuda} &= 12.4 \\
\text{cudnn} &= 9.1.0 \\
\text{pytorch} &= 2.4.0 \\
\text{torchvision} &= 0.19.0 \\
\text{python} &= 3.10.16
\end{aligned}
\tag{31}
$$

The third-party library used in our study has also been attached to the package. Users are encouraged to run the code in the same setting. For other possible environments, the performance reproduction results may not be guaranteed to be the same.

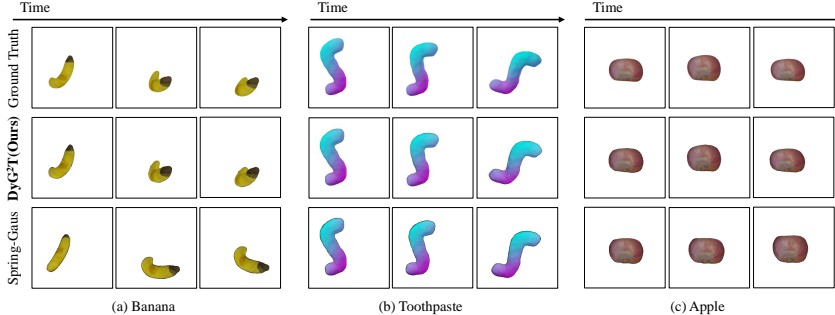

Figure 4: Qualitative results of dynamic reconstruction for DyG$^2$T and baselines. (a), (b), and (c) correspond to Banana, Toothpaste, and Apple, respectively.

Table 10: Quantitative results of dynamic reconstruction for DyG$^2$T and its variants. "-" indicates memory overflow during metric computation caused by the explosion of point cloud size. "Num PT" denotes the number of particles in the reconstructed point cloud.

| Variants | CD↓ | EMD↓ | PSNR↑ | SSIM↑ | LPIPS↓ | Num PT |
|---|---|---|---|---|---|---|
| Overweight | 0.008 | - | 32.027 | 0.981 | 0.014 | 311198 |
| Unequal | 0.007 | 0.002 | 33.038 | 0.980 | 0.009 | 12691 |
| Large Ner | 0.008 | 0.001 | 33.744 | 0.985 | 0.007 | 7566 |
| Small Ner | 0.010 | 0.003 | 32.753 | 0.981 | 0.013 | 7564 |
| Ours | **0.008** | **0.001** | **34.001** | **0.985** | **0.007** | 7307 |

## C ADDITIONAL RESULTS

### C.1 TRACKABLE PARTICLE-BASED REPRESENTATION BY DYNAMIC RECONSTRUCTION

#### C.1.1 QUALITATIVE AND QUANTITATIVE RESULTS OF DYNAMIC RECONSTRUCTION

In Section 4.2 of the main text, we present the quantitative results of dynamic reconstruction for DyG$^2$T and the baselines on the synthetic Spring-Gaus dataset Zhong et al. (2024). The complete quantitative results compared with more baselines are shown in Table 9. Some results are cited from Spring-Gaus Zhong et al. (2024). Furthermore, we present the qualitative results of this task in Figure 4. As shown in Figure 4(a), Spring-Gaus Zhong et al. (2024) exhibits significant spatial mismatch during dynamic reconstruction under the supervision of visible frames. Figure 4(c) shows another manifestation of reconstruction error: inaccurate motion estimation leads to incorrect judgments about the object's contact timing and rebound amplitude, resulting in biased object appearances. The reconstruction deviations shown in Figures 4(a) and (c) may mislead dynamics modeling and cause severe error accumulation. Moreover, as shown in Figure 4(b), Spring-Gaus performs relatively well on the Cross object, suggesting that its reconstruction performance is sensitive to object appearance and attributes, revealing a limitation in generalization compatibility.

#### C.1.2 PHYSICAL PRIORS ON THE DYNAMIC RECONSTRUCTION

To further investigate the impact of different physical priors on the dynamic reconstruction performance of DyG$^2$T, we design four variants: Overweight, which applies overly strong physical constraint weights inspired by GS-Dynamics Zhang et al. (2025a); Unequal, which assigns imbalanced weights to 2D rendered images and semantic segmentation maps; and Large/Small Ner, where the number of Gaussian neighbors $k_{Ph}$ is set to 20 and 3, respectively. The quantitative results are reported in Table 10.

In our experiments, the optimal dynamic tracking parameters achieve the best description of object motion with the smallest point cloud scale. Although the Unequal and Large Ner variants achieve comparable point cloud quality (CD & EMD) to our full model, they suffer from noisy outlier points in the reconstructed clouds (as shown in Figure 5), which directly degrade the 2D rendering performance—PSNR for Large Ner; PSNR, SSIM, and LPIPS for Unequal. We attribute this to two factors: the imbalanced weighting of semantic segmentation maps (Unequal), and the overly

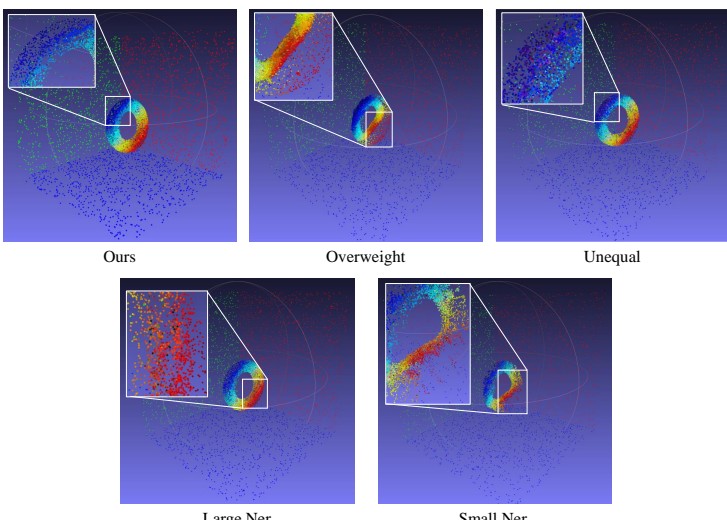

Figure 5: Visualization of dynamic reconstruction point clouds generated by DyG²T and its variants. Zoomed-in views are provided to highlight the fine-grained structural details of the reconstructed point clouds. The background planes composed of blue, red, and green points represent the planes $z = 0$, $x = 0$, and $y = 0$, respectively.

Table 11: Ablation study on using different frames as the alignment reference. "Align Refer" indicates the index of the selected reference frame.

| Align Refer | CD↓ | EMD↓ | PSNR↑ | SSIM↑ | LPIPS↓ |
|---|---|---|---|---|---|
| t-2 | 0.250 | 0.125 | 16.810 | 0.882 | 0.211 |
| t-1 (Ours) | **0.055** | **0.029** | **17.904** | **0.919** | **0.122** |
| t | 0.131 | 0.068 | 16.964 | 0.893 | 0.175 |

broad Gaussian neighborhoods (Large Ner), both of which introduce excessive noise into dynamic reconstruction.

In contrast, the Overweight variant produces an excessively large point cloud (311,198 particles) due to overly strong physical constraints. This results in distorted and deformed point clouds that are difficult to evaluate quantitatively (EMD computation leads to out-of-memory errors), and also exhibit degraded 2D rendering quality. The Small Ner variant, which considers only 3 Gaussian neighbors, yields the worst point cloud quality; the visualizations in Figure 5 further support this finding. We believe that enforcing physical constraints within such a limited local neighborhood leads to a disconnect between local and global structures, ultimately hampering coherent dynamic reconstruction.

## C.2 SPATIAL-TEMPORAL FEATURE COMPLETION AND AGGREGATION

In the Object-level Dynamic Temporal Aggregation module, we adopt the Motion Align Net to achieve flexible alignment of cross-frame motion, reducing trajectory prediction errors. In practice, Motion Align Net requires selecting a reference frame to compute relative motion offsets. To investigate the impact of different reference frames on dynamics modeling, we conduct ablation studies by varying the reference frame used for alignment.

As shown in Table 11, variants that use either the $t - 2$ or $t$ frame as the alignment reference fail to achieve accurate trajectory prediction. These two frames represent the endpoints of the visible motion observation window. When using either as the reference, alignment must span across the intermediate frame. For instance, if the $t - 2$ frame is selected as the reference, aligning the $t$ frame requires skipping over the $t - 1$ frame. We argue that this skip-frame alignment increases the difficulty of computing reliable cross-frame offsets, making motion compensation more challenging. Therefore, in practice, we adopt the $t - 1$ frame as the reference frame to balance observability and alignment stability.

Table 12: Quantitative results of DyG$^2$T and its GNN variants on the dynamics modeling of the Heterogeneous Torus.

| Methods | CD↓ | EMD↓ | PSNR↑ | SSIM↑ | LPIPS↓ |
|---|---|---|---|---|---|
| DyG$^2$T-GNN | 0.021 | 0.008 | 13.995 | 0.891 | 0.143 |
| DyG$^2$T(Ours) | **0.015** | **0.000** | **14.080** | **0.893** | **0.129** |

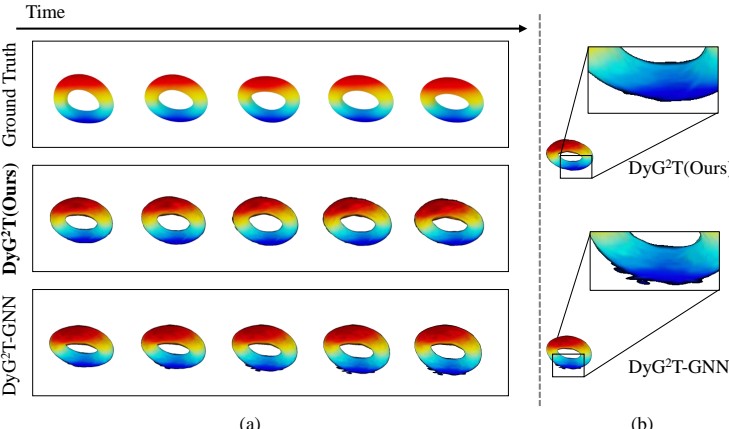

(a)     (b)

Figure 6: (a) Qualitative results of DyG$^2$T and its GNN variants on the dynamics modeling of the Heterogeneous Torus. (b) Local zoom-in views for evaluating the fine-grained details of dynamics modeling.

## C.3  DYNAMICS MODELING AND REASONING OF MOVING OBJECTS

We further present the dynamics modeling results of DyG$^2$T and its GNN-based variant on the Heterogeneous Torus. In the DyG$^2$T-GNN variant, we retain the Spatial-Temporal Features Completion and Aggregation mechanism while replacing the Particle Graph Transformer module with a GNN. The quantitative results are shown in Table 12. Although DyG$^2$T-GNN benefits from sufficient spatiotemporal semantic information via the hierarchical attention mechanism, it struggles to accurately capture the complex internal force propagation patterns of heterogeneous materials due to limitations such as over-smoothing Sun et al. (2022; 2025); Li et al. (2019a). This significantly hampers dynamics reasoning, leading to degraded performance in both 3D point cloud reconstruction (as measured by CD and EMD) and 2D rendering (PSNR, SSIM, and LPIPS), compared to DyG$^2$T.

The qualitative results in Figure 6 further support these observations. The heterogeneous material properties of the Torus increase the difficulty of dynamics modeling. Without the capacity to globally capture multi-scale force propagation, DyG$^2$T-GNN exhibits noticeable error accumulation during the later stages of dynamics reasoning (Figure 6(a)). This results in evident artifacts in the inferred appearance of the Heterogeneous Torus, as shown in Figure 6(b).

