# OpenReview forum: "DyG$^2$T: Modeling Object Dynamics with 3D Gaussian Temporal-Spatial Particle Graph Transformer"
_ICLR.cc/2026/Conference — ICLR 2026 Conference Withdrawn Submission_

### Official Review · Reviewer_uSDg · 2025-10-28

**Soundness:** 3
**Presentation:** 3
**Contribution:** 3
**Rating:** 6
**Confidence:** 3

**Summary:**

This paper addresses a critical gap in dynamic object modeling—accurate trajectory prediction and fine-grained dynamics capture from limited observations—and delivers a technically rigorous framework with compelling experimental validation. Its contributions are well-aligned with the needs of embodied AI, robotics, and dynamic scene understanding, making it a valuable addition to the field.

**Strengths:**

The paper introduces several novel and well-integrated components that distinguish it from existing work:
1. Unlike prior methods (e.g., GS-Dynamics, Spring-Gaus) that rely on downsampled Key Points alone, DyG²T enriches Key Point features with two critical cues: (1) fine-grained edge information from raw 3D Gaussian particles via PointNet-based neighborhood aggregation, and (2) inter-frame motion offsets via Motion Align Net (MAN). This addresses the longstanding issue of "feature homogenization" in sparse Key Point modeling.
2. The replacement of vanilla GNNs with a Transformer-based graph module is a key innovation. By establishing direct force propagation paths between contact-near and distant Key Points, it preserves discriminative long-range dependencies— a limitation of GNNs, which only model local neighborhood interactions.
3.The framework tightly couples 3D Gaussian-based dynamic reconstruction (building on Dyn3DGS) with physics-aware dynamics reasoning, avoiding the "reconstruction-to-modeling disconnect" in prior pipelines that treat these steps as separate.

**Weaknesses:**

1. Limited evaluation of heterogeneous materials. The Unity3D-H dataset only includes a single polyhedron with two materials, and the Heterogeneous Torus experiment (Table 4) focuses on a single object type. This limits the authors’ claim of "strong cross-object generalization" for heterogeneous materials.
2. Lack of computational efficiency analysis. While the paper mentions DyG²T has "higher computational efficiency" (Section 4.3), it provides no quantitative data to support this.
3. Ambiguity in Motion Align Net (MAN) Design. The paper describes MAN as "compensating inter-frame motion offsets" (Section 3.2.2) but provides insufficient details on its architecture: The mapping from cross-frame features XInC​ to motion offsets δt−2 and δt is only briefly mentioned (via tanh constraint) but not fully defined.

**Questions:**

1. The Particle Graph Transformer uses "Gated Residual" to mitigate over-smoothing (Section 3.3), but the Appendix does not provide the full equation for this residual connection. Could you explicitly define how the gate weighs the Transformer output and residual?
2. In the noisy input experiment (Table 4), DyG2Tnoisy​ uses "perturbations to the central trajectory"—what is the magnitude and type of perturbation (e.g., Gaussian noise, random translation)? How does this compare to real-world noise (e.g., sensor noise in RGB cameras)?

---

### Official Review · Reviewer_tFV2 · 2025-10-29

**Soundness:** 2
**Presentation:** 2
**Contribution:** 2
**Rating:** 4
**Confidence:** 4

**Summary:**

This paper addresses critical limitations in existing object dynamics modeling, where reliance on downsampled key points and local neighborhood interactions leads to a "loss of fine-grained details" and "homogenized particle representations". The authors propose a novel framework, DyG$^{2}$T, which leverages "spatiotemporally completed particle representations" and a "Particle Graph Transformer" to overcome these issues. This approach is designed to model "multi-scale force propagation" and capture "discriminative long-range dependencies" often missed by traditional GNNs. Extensive experiments on both synthetic and real-world datasets demonstrate that the DyG$^{2}$T framework achieves significantly more accurate "trajectory decoding" and exhibits strong "cross-object generalization" capabilities compared to current baselines.

**Strengths:**

- The paper addresses a timely and significant problem: accurately modeling object dynamics from visual observations. This is a fundamental challenge for embodied AI and physical world interaction.
- The experimental validation is thorough. The authors test their method on multiple synthetic and real-world benchmarks, including a self-constructed "Heterogeneous" dataset to test generalization.
- The proposed DyG$^{2}$T framework demonstrates superior performance over baselines across all key 3D and 2D metrics.

**Weaknesses:**

1. The paper lacks consistent punctuation (e.g., commas or periods) after display equations, which detracts slightly from the professional presentation.
2. The justification for the Multi-head Position-Aware Attention (Eq. 2 & 3) appears conceptually flawed. The authors state this is a cross-space feature fusion, but the implementation is contradictory. Both the Query (Q) and Key (K) are derived from the same source ($X_{CoP}$), making the attention score calculation ($\text{softmax}(Q \cdot K^T)$) a self-attention mechanism on the coordinate features. This self-attention score is then used to aggregate values (V) from a different source ($X_{Po}$). This is not a standard cross-attention (where K and V are from the same source, and Q is from another). This unconventional design requires a much stronger theoretical and empirical justification, as it is not clear why a self-attention score on coordinates is the appropriate way to weight neighborhood features.
3. The "Motion Align Net (MAN)" module is not sufficiently justified. The claim that it learns inter-frame motion offsets is not substantiated beyond the module's description. It is unclear how processing concatenated features from three frames (Eq. 5) guarantees meaningful alignment. The authors should provide further analysis, such as a qualitative visualization or a case study or ablation study, to demonstrate that the MAN module is indeed learning effective alignment rather than just acting as an additional learnable transformation.
4. The experimental setup introduces a significant confounding variable. The authors use an existing method (Dyn3DGS) for dynamic reconstruction and show in Table 1 that their implementation outperforms the baseline's input. Since this reconstruction phase is not a novel contribution of this paper, it is difficult to ascertain whether the performance gains in Table 2 stem from the proposed DyG$^{2}$T architecture or simply from the higher-quality input. A more convincing comparison would involve providing this superior reconstruction data to all baseline models (e.g., GS-Dynamics) and re-evaluating their performance.
5. The paper lacks a crucial comparison of model complexity. By replacing a GNN with a "Particle Graph Transformer" and introducing additional modules like MAN, the parameter count of DyG$^{2}$T is presumably much larger than the baselines. The authors should report the parameter counts for all compared methods to clarify if the performance gains are a result of a more effective architecture or simply a much larger model.

**Questions:**

See the critical points raised in the Weaknesses.

---

### Official Review · Reviewer_Mk9p · 2025-10-30

**Soundness:** 2
**Presentation:** 1
**Contribution:** 2
**Rating:** 2
**Confidence:** 4

**Summary:**

This paper proposes a framework, DyG2T, to model dynamic scenes as a particle graph transformer. Spatially, it leverages the raw point coordinates and relative position information from the neighbourhood to enrich the key point features. Temporally, it estimates the relative motion offset of each particle between adjacent frames. Lastly, the graph transformer model particle interactions globally and estimates future motions. In experiments, DyG2T surpasses baselines in terms of dynamic reconstruction and motion prediction on Spring-Gaus and Unity3D-H datasets.

**Strengths:**

1. Analysis of experimental results is comprehensive.

2. Ablation studies for different modules are conducted thoroughly.

**Weaknesses:**

1. The writing and presentation of this paper are very hard to follow:

(i) It is suggested to use commonly adopted terms instead of introducing new ones overwhelmingly (e.g., particle-based appearance descriptors and initial spatial representations can be simplified as 3D Gaussian parameters).

(ii) It is suggested to avoid plain descriptions of unimportant implementation details, but focus on high-level design choices.

2. Some highly relevant baselines are not discussed and compared:

(i) GIC: Gaussian-informed continuum for physical property identification and simulation (NeurIPS 2024).

(ii) FreeGave: 3D Physics Learning from Dynamic Videos by Gaussian Velocity (CVPR 2025).

3. Designs for some modules are not well-justified:

(i) What is the relative motion supplement?  Why is it necessary to compute it?

(ii) Why is Motion Align Net (MAN) considered an object-level dynamic temporal aggregation, since it is still performed on key points?

(iii) Why are k-nearest neighbors aggregated through PointNet believed to be able to capture ``fine-grained edge features”?

4. Self-collected Unity3D-H datasets are not well elaborated and presented.

**Questions:**

1. DyG2T uses frame-by-frame optimization to get one-to-one correspondence of particles across frames. Is this process time-consuming? When extracting key points from raw particles, how to guarantee the correspondence?

2. Since temporal aggregation is only performed between three adjacent frames, can DyG2T capture long-term motion?

3. There are many important hyperparameters in DyG2T, such as the number of neighbors, the number of key points, and the number of edges. From ablation studies, the choice of these hyperparameters can largely affect final predictions. Given that the experiments are only conducted on two datasets, are those hyperparameters carefully tuned? How well does this setting generalize to video data in other domains?

---

### Official Review · Reviewer_8NkT · 2025-10-31

**Soundness:** 2
**Presentation:** 1
**Contribution:** 2
**Rating:** 2
**Confidence:** 2

**Summary:**

This paper tries to use a series of transformer models to fit the hidden physics rules in a dynamic reconstruction framework from given observed multi-view videos. The main focus of their pipeline is to model how the implicit forces are propogated between particles in different scale. By achieving this, they show good future prediction performance on three datasets.

**Strengths:**

1. The paper has a detailed explanation in implementation.
2. Thorough ablation studies on each modules, giving readers’ an in-detail understanding about the ability for each modules.
3. The research problem is important, because enabling models to understand physics is beneficial for many downstream tasks such as robotics and the world model.

**Weaknesses:**

Although this paper achieves great performance in given datasets compared to the baselines, there are strong weaknesses of the proposed methods.
1. Unlike spring-gs and pac-nerf fitting parameters for simulation pipelines, this paper uses transformers to implicitly learn the propagation rule of forces. However, this learned propagation rule is only fitted per-scene-wise, which means it could only overfit to observed forces. How the model can be generalized to novel environment setting and unobserved external forces propagation is NOT evaluated and shown. Although the future prediction to some extent requires this ability, the future parts in the dataset is quite simple, and the configurations of the objects are not `new' for the network, it is still not clear whether the model really learns physics. Therefore, in order to really support the arguments that the model learns the physics, experiments about resimulation with different initial object configurations and differnet environment settings are important.
2. The baselines and related works are simple. There are missing future prediction methods to discuss, such as FreeGave (Li et al, CVPR2025), GaussianPrediction (Zhao et al, SIGGRAPH2024). As for the baselines to directly compare with, GIC (Cai et al, NeurIPS 2024 oral) should be included.
3. The presentation is too tedious. There are too many efforts put in the implementation part (the definition for every single mlps and transformers), making it extremely hard for people to understand what is the main purpose. After reading all the equations with a great effort, I finally find most of the equations are not important. I strongly request the authors to revise the method part.

I’m open to increase my score if the authors can address my concerns above.

**Questions:**

1. Do you include the background in your experiments? Only foreground objects are shown in the qualitative results.
2. The visual metrics, such as PSNRs, SSIMs, LPIPSs, in Spring-Gaus synthetic dataset is extremely low. Even the best 14.080 PSNR is far worse than expectation, can the authors explain the too low PSNRs? And the scores for spring-gaus is too low compared to its original paper. Can the authors explain this?

---

### Official Review · Reviewer_17kd · 2025-11-07

**Soundness:** 2
**Presentation:** 2
**Contribution:** 2
**Rating:** 4
**Confidence:** 2

**Summary:**

This paper presents DYG2T, a framework that employs spatiotepmorally completed particle representations to support multi-scale force propagation modeling. In particular, particle-based representations are learned through dynamic reconstruction. After the spatial-temporal feature completion and aggregation steps, a particle graph transformer is used for modeling the dynamics. The proposed method has been evaluated on Spring-Gaus synthetic dataset and self-curated Unity3D-H dataset. Experimental results show performance improvements over the previous state of the art methods.

**Strengths:**

1.[effectiveness] The proposed method clearly outperforms the previous state of the art methods on all datasets in terms of the numerical metrics.

2.[clarity] ablation studies have been carried out to demonstrate the effectiveness of some modules in the proposed method.

**Weaknesses:**

1.[efficiency] The proposed method takes fine-grained details into consideration (L017). Does such a design incur more computational complexity or space complexity? Especially compared with the previous methods.

2.[typesetting] Please use "\citep" or "\citet" properly to avoid messing up the text with the pain "\cite". The messed-up text is not smooth to read and could have been avoided if the authors had done proofreading.

3.[ablation] The \epsilon paramerer for the loss funciton is not discussed. How does the selection of this parameter impact the model behavior?

4.[clarity] The proposed method seems to introduce less performance improvements over the previous state of the art methods on the Unity3D-H dataset, numerically in Table 3. Is that because the polyhedron (L307) is much simpler than other shapes (e.g., Dog)? Or any other reason?

5.[clarity] This paper introduces too many modules which is not quite easy to follow.

**Questions:**

See weaknesses.

---

### Note · Authors · 2025-12-02

I have read and agree with the venue's withdrawal policy on behalf of myself and my co-authors.